# Effects of Nano TiC on the Microhardness and Friction Properties of Laser Powder Bed Fusing Printed M2 High Speed Steel

**Yan Liu, Dingguo Zhao \*, Yue Li and Shuhuan Wang**

College of Metallurgy and Energy, North China University of Science and Technology, Tangshan 063009, China; liuyan1342991123@163.com (Y.L.); liyue1593126@163.com (Y.L.); wshh88@ncst.edu.cn (S.W.)
\* Correspondence: zhaodingguo@ncst.edu.cn

**Abstract:** In this work, TiC/M2 high speed steel metal matrix composites (MMCs) were prepared using the ball milling method and laser powder bed fusing process. By controlling the TiC content in TiC/M2HSS, the grain size, phase composition, and frictional wear properties of the samples were enhanced. The results showed that when TiC/M2HSS was supplemented with 1% TiC, the surface microhardness of the samples increased to a maximum value and the wear volume decreased by approximately 39%, compared to pure M2HSS. The hardness and friction wear properties of the TiC/M2HSS composites showed a decreasing trend as the TiC content increased, owing to an increase in internal defects in the samples, as a result of excess TiC addition. The physical phases of the TiC/M2HSS MMC samples prepared by LPBF were dominated by the BCC phase, with some residual FCC phases and carbide phases. This work explored the possibility of enhancing the frictional wear performance of TiC/M2HSS samples by controlling the TiC content.

**Keywords:** M2HSS; LPBF; Namo-TiC; friction properties

## 1. Introduction

W. Breelor developed M2 high speed steel high carbon content tool steel developed in 1937, which has been widely used in machining as a lathe sharpening material owing to its good wear resistance [1]. At present, the main production processes consist of melting and casting, electroslag remelting, and powder metallurgy. The traditional production method is mainly based on casting; however, its high alloy content will result in the solidification process of tissue segregation and carbide precipitation, resulting in the subsequent forging process of high-speed steel cracking, limiting the application of high-speed steel [2]. In the 1960s, researchers developed powder metallurgy techniques to improve the phenomenon of carbide segregation. However, M2 HSS prepared by powder metallurgy still requires complex machining to produce parts, making it difficult to machine complex structural parts such as ultra-thin-walled internal cavities. Thus, the further development of machining and forming techniques for M2 HSS is required.

Laser powder bed fusion (LPBF) technology consists of a metal additive manufacturing technique that can be used to prepare workpieces with complex geometries and fine grain structures [3–5]. In the LPBF process, the metal powder is laid flat on a powder bed and rapidly melted and solidified by a high-speed laser sweep, with melting and solidification occurring within $10^6$–$10^7$ s [6–10]. Kempen et al. [11] successfully improved the warpage and cracking of SLM-prepared HSS by preheating the substrate temperature and investigated the microstructure of M2 HSS prepared by the SLM process. Liu et al. [12] demonstrated that the solidification process of the HSS melt pool was significantly higher in the center of the melt pool than at the edges, according to MATLAB simulations. The solidification of the melt pool in the center was found to be longer than the edge temperature, which was also conducive to the growth of crystals. Karolien. Kempen et al. [13]

prepared HSS lumps with a density of 98% using a substrate preheating temperature of 200 °C. Thus, the above researchers conducted a comprehensive study of the processing methods, microstructure, and physical phases of M2 HSS in the LPBF process.

Metal matrix composites (MMCs) have received widespread attention due to their excellent frictional wear properties [14–24], and TiC [25] is a very stable compound that has been commonly added to metal matrices to increase the wear resistance of composites due to its high melting point and strong chemical stability [26]. Li et al. [20] prepared TiC-containing 316 L stainless steel by LPBF and found that the addition of TiC refined the grain size of the sample, thus increasing sample strength. In addition, Luo et al. [16] used LPBF to prepare TiC-added magnesium alloys, which significantly strengthened the wear resistance of the magnesium alloy. However, no studies have been conducted on using LPBF to prepare MMCs with M2HSS as the matrix. High speed steel has mainly been used as a tool for machining equipment; therefore, it requires good wear resistance to enhance service life. In this study, we used LPBF to prepare TiC/M2HSS composites with different contents, to investigate the feasibility of using the LPBF process for preparing formed crack-free high density TiC/M2HSS composites, and we investigated the TiC/M2HSS material through microscopic characterization and friction wear performance testing.

## 2. Materials and Experiments

### 2.1. Preparation of Materials

The metal powders used for LPBF (range 15–53 μm) were obtained from M2 HSS ingots by Ar gas atomization and the powder chemical composition was determined by XRF (XRF-1800, Scimadzu, Kyoto, Japan), as listed in Table 1, and Learning ManagementSystem (LMS, Brno, Czech Republic), as shown in Figure 1a. The particle size distribution of the powders was determined using a laser diffraction particle size distribution meter and the results are shown in Figure 1b. The powder was mostly spherical with a Dv50 of 30.5 μm.

Irregularly shaped TiC nanopowders with an average size of 100–500 nm (purity of no less than 99%) and M2 HSS aerosolized powder were used as raw materials. The TiC nanopowders were added to the M2HSS powder in proportions of 1%, 3%, and 5%, and then ball milled and mixed in an Ar atmosphere using a planetary ball mill. The ball to powder weight ratio was 5:1, the ball mill speed was 240 r/min, and the ball milling time was set to 4 h, where the powder was cooled after each hour of ball milling for 15 min. After ball milling, the TiC nanopowders were uniformly distributed on the surface of the M2HSS powder without any obvious agglomeration. An energy dispersive X-ray spectroscopy (EDS, Oxford Swift 3000, Abingdon, UK) energy spectral surface scan of the prepared composite powder was obtained, as shown in Figure 1c, with different colors for Fe, Ti, and C, which showed that the TiC particles were uniformly distributed on the powder surface. The powders were preheated at an atmosphere temperature of 80 °C for 10 h to enhance flowability before LPBF.

**Table 1.** Chemical composition of the M2HSS powder (wt.%).

| Composition | W | Mo | Cr | V | C | Mn | Fe |
|---|---|---|---|---|---|---|---|
| Nominal composition (wt.%) | 5.7–6.0 | 4.6–4.9 | 3.6–3.9 | 1.6–1.9 | 0.7–0.9 | 0.1–0.3 | Bal. |
| Actual composition (wt.%) | 5.87 | 4.93 | 3.87 | 1.89 | 0.84 | 0.29 | Bal. |

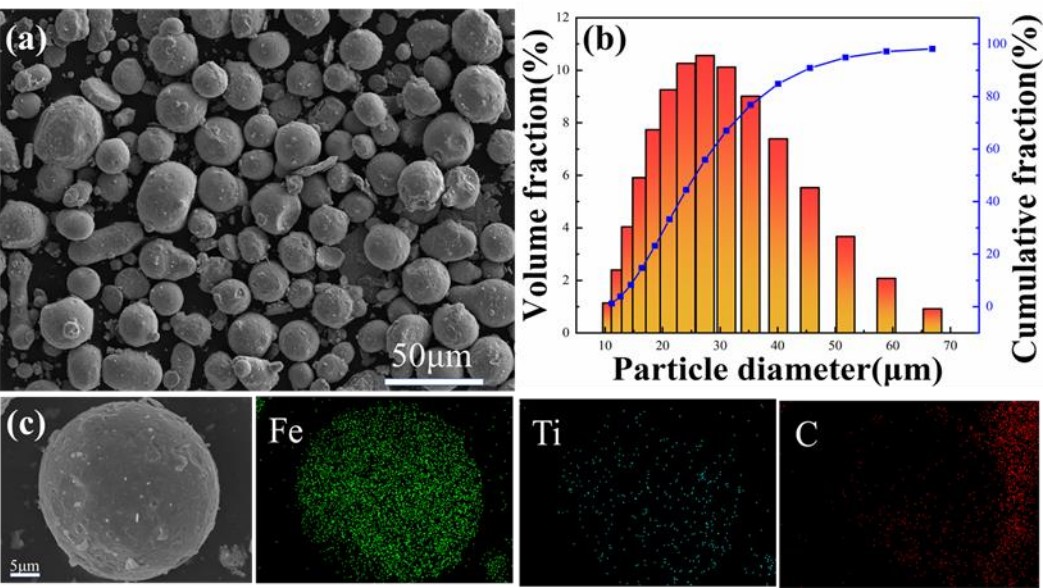

**Figure 1.** (**a**) Morphology of the TiC/M2HSS powder; (**b**) particle size distribution of the M2HSS powder; (**c**) elemental distribution of Fe, Ti, and C on a single M2 powder particle.

### 2.2. LPBF Process

All samples were LPBF-printed using laboratory-built equipment equipped with a ytterbium continuous single-mode fiber laser (maximum power 500 W, wavelength 1080 ± 5 nm) and focusing optical system using a 254 mm focal length F-θ lens, which produced a focused beam with a spot diameter of approximately 100 μm. The laboratory equipped the equipment with a substrate pre-heating device that allowed the stainless-steel substrate to be pre-heated up to 300 °C. During the LPBF process, Ar gas was used as the protective gas to prevent oxidation and the oxygen content in the forming cavity was below 100 ppm.

Figure 2a shows a schematic of the LPBF process, where a 67° oblique zoning scanning strategy was used between the adjacent layers in the forming process to reduce the thermal stresses [27], as shown in Figure 2b, where each zone was spaced 4 mm apart, using a laser to scan the adjacent zones. The printed samples are shown in Figure 2c, where two cubes, of 8 mm × 8 mm × 6 mm in size were used for microstructure, phase identification, and hardness testing, and a 15 mm × 15 mm × 5 mm sample was used for frictional wear testing, which was cut from the substrate after machining using electrical discharge machining (EDM, Jinli, Taizhou, China).

### 2.3. Characterization and Mechanical Testing

The densities of the samples were determined according to the Archimedes drainage method. The samples were polished with 240 to 5000 grit sandpaper and polished with 1 μm diamond polish for 20 min before they were observed under an optical microscope and by scanning electron microscope (SEM, Quanta 650 FEG, FEI, Madison, WI, USA) for defects such as holes and cracks.

For observations, the substrate was preheated at 300 °C, the scanning pitch (h) was fixed at 100 μm, the layer thickness was 30 μm, and the laser power values were 210, 240, 270, and 300 W, with scanning speeds of 500, 600, 700, and 800 mm/s, respectively. LPBF used $\eta$ volumetric energy density as the evaluation criterion for the printing parameters [28]. The volumetric energy density $\eta$ (J/mm$^3$) could be calculated by

$$\eta = \frac{p}{v\mathrm{hd}} \tag{1}$$

where $p$ denotes the laser power, $v$ is the scanning speed, h is the scanning interval, and d is the powder layer thickness.

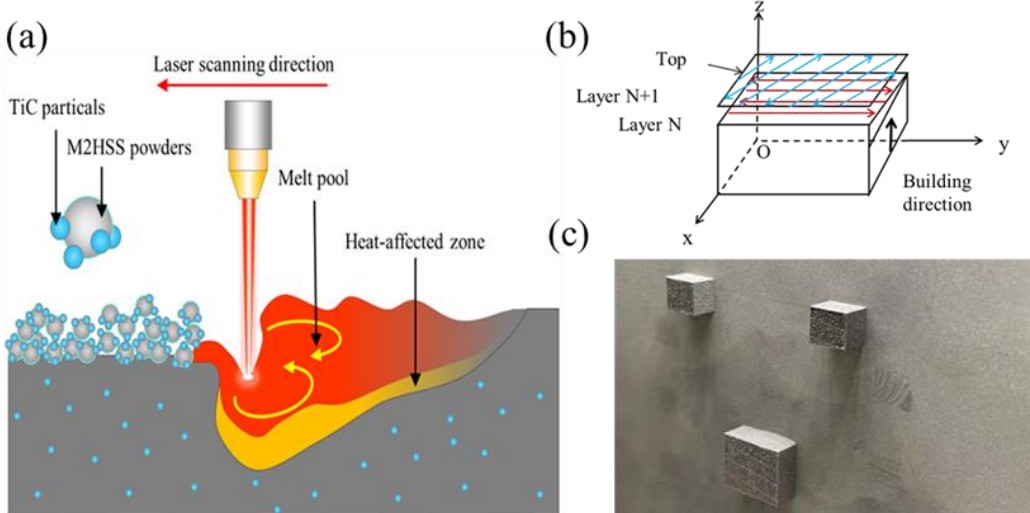

**Figure 2.** Schematic (**a**) diagram of LPBF TiC/M2HSS composite processing; (**b**) laser scanning strategy; (**c**) TiC/M2HSS samples printed by LPBF.

The samples were nano-indented using a Bruker TI 980 (Bruker Nano Surfaces, Madison, WI, USA) to obtain the micro-zone hardness and modulus of elasticity of the samples [29,30], using a Berkovich probe (Center for Tribology, Madison, WI, USA) and an experimental fixation depth of 1000 nm.

The relationship between the density and bulk energy density for 1% TiC/M2 is shown in Figure 3, which was divided into three regions based on the magnitude of the volumetric energy density, and typical light microscopy photographs, which were inserted in each region. The best combination of process parameters was obtained in region 2, due to the presence of pores and cracks on the surface of region 1, the presence of a large number of cracks on the surface of region 3, and the low number of defects on the surface of region 2. In this study, the optimal parameters were chosen as the laser power and scanning speed of 270 W and 700 mm/s, respectively. Figure 3 shows the relationship between density and volumetric energy density.

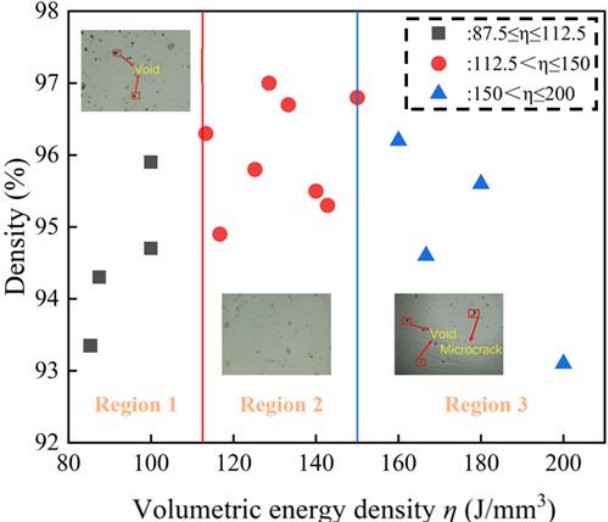

**Figure 3.** Relationship between density and the volumetric energy density.

The samples were subjected to metallographic etching using a composition of $HNO_3$ (5 mL) and anhydrous ethanol (45 mL). The surfaces of the samples were wiped with the etching solution for 180 s and rinsed clean using anhydrous ethanol, and the microstructure and wear morphology of the samples were observed under an SEM. The samples were scanned to determine their crystal structures using a Rigaku SmartLab SE (Japan) X-ray diffractometer (CuKα radiation, 40 KV, 40 mA, RIGAKU, Tokyo, Japan) with a scanning angle range of 10° to 80° and a scanning step angle of 0.013° for 1 s each time.

The friction and wear experiments were conducted using a Bruker (CETR, Center for Tribology, Wellesley, MA, USA) UMT-2 friction and wear tester with 6 mm Si3N4 ceramic balls, a normal load of 60 N, a friction stroke of 10 mm, and a sliding rate of 10 mm/s for 60 min to obtain the friction factor-time curves for the different samples. A laser confocal microscope (LSCM) (VK-X1000 of Keyence Corporation, Tokyo, Japan) with a wavelength of 661 nm was used to photograph the wear marks of the samples and to obtain the three-dimensional profiles and two-dimensional curves of the wear volume and wear marks [31,32]. The areas after the friction test were photographed using the SEM to analyze the frictional wear process. The hardness values of the polished samples were measured using a micro hardness tester (FM-800, FUTURE-TECH CORP, Toyko, Japan) at room temperature with a fixed loading force of 1 kg and a loading time of 15 s. Five randomly selected points on the surface of each sample were tested and the average value was taken as the hardness value of the sample.

## 3. Results and Analysis

### 3.1. TiC Action and Micro-Morphology Defects

Figure 4 shows the top surface of the LPBF-printed sample as observed under an optical microscope. Sample A0 had a smoother surface quality with no large holes or cracks, while sample A1 had micro cracks on the surface and grey precipitates were observed on the surfaces of some samples (magnified view). The precipitates were caused by the agglomeration of TiC in the matrix due to the addition of 1% TiC to sample A1 [33]. Sample A3 contained some circular holes on the surface of the sample, which were possibly caused by gas in the metal powder or the high purity argon atmosphere during the LPBF process [34]. Sample A5 had more holes distributed on the surface and the grey TiC precipitates were significantly denser (enlarged image), which was caused by the addition of more TiC and the fact that TiC in the melt pool reduced the viscosity of the melt pool [19], causing more defects on the surface and easier agglomeration.

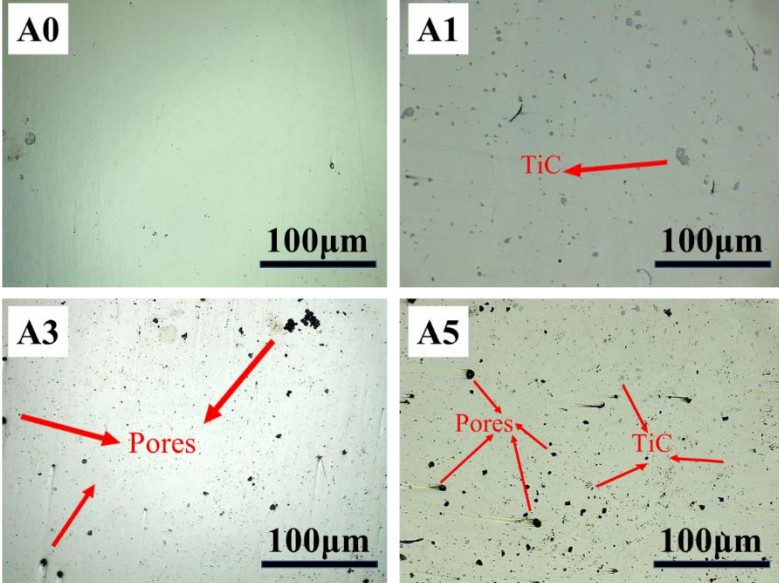

**Figure 4.** OM morphologies of the LPBF-printed TiC/M2HSS.

Figure 5a–d shows the different types of defects on the top surface of a selected A5 sample with 5% TiC addition, which was photographed using an SEM. Figure 5a shows a typical crack-like defect where the sample crack spread around along the unfused hole, eventually causing the sample surface to lose hardness. Figure 5b shows the crater-like defect and presence of TiC particles around the crater. This was due to the high viscosity of the melt, which allowed the melt to solidify without completely covering the crater. The crater shown in Figure 5c contained some spherical unmelted particles and a large amount of TiC around the crater, which was not enough to melt all of the powder due to the high energy absorption of TiC by the matrix. As the LPBF was built up layer by layer, defects were covered by a new layer when present, as shown in Figure 5d. The cause of these defects was generally considered to be due to the hole locking effect of the melt pool splash and metal vapor recoil [19,35].

During the LPBF-forming process when the laser irradiated the surface of the powder bed, micro-melt pools formed instantaneously, which overlapped together to form a dense structure, and the following relationship formed between the liquid phase viscosity $\mu$ of the pool, the pool temperature $T$, and surface tension $\gamma$ [35]:

$$\mu = \frac{16}{15}\sqrt{\frac{m}{k_B T}}\gamma \tag{2}$$

where $m$ denotes the atomic mass; $k_B$ is the Boltzmann constant, and $\gamma$ is the surface tension of the liquid phase.

The liquid phase viscosity $\mu$ of the melt pool was proportional to $\gamma$ and inversely proportional to the quadratic root of the melt pool temperature $T$ when m and $k_B$ were constant; thus, the melt pool viscosity decreased with an increasing temperature $T$ when $\gamma$ was constant. A moderate reduction in the melt pool viscosity was beneficial for improving the spatter and surface topography during LPBF; however, too low of a melt pool viscosity would cause the melt pool to become unstable and increase spatter. Owing to the high melting point of the titanium carbide nanoparticles, they were difficult to decompose during LPBF, and could only be retained in the melt pool as particles and the increase in the viscosity of the liquid phase $\mu$, reduction in the liquid metal flow, and the melt pool solidification rate during LPBF were extremely fast ($10^6$–$10^7$ k/s) [36]. Therefore, a higher energy input was required to increase the melt temperature and reduce the melt viscosity to increase the wetting of the melt. As the energy input gradually increased, the density of the LPBF molded portion was increased to a certain extent. However, when the energy input was too high, it caused the powder bed to absorb too much energy and although the melt pool surface tension $\gamma$ decreased and the melt pool viscosity $\mu$ also decreased, the thermal stress generation and microcrack expansion led to a decrease in the densities of the finished products [37,38]. These observations were consistent with the results shown in Figure 3.

The differences in laser absorption between M2HSS and the TiC powders resulted in a large temperature gradient inside the micro-melt pool, which in turn induced a surface tension gradient and Marangoni convection, promoting the flow of the liquid phase and generating capillary tension [39]. Subsequently, when applied to the irregularly shaped TiC particles, this promoted their movement and redistribution in the pool. At high levels of TiC particles, the particles collided with each other and agglomerated, forming larger particles at the micron scale. The excessive increase in TiC particles also led to an increase in the viscosity of the melt pool and a decrease in fluidity. Severe spherification, which reacted to the macroscopic level by the presence of a large number of spherical holes on the polished surface, was a phenomenon consistent with the observations made by Gu et al. in the preparation of TiC/Ti composites [40]

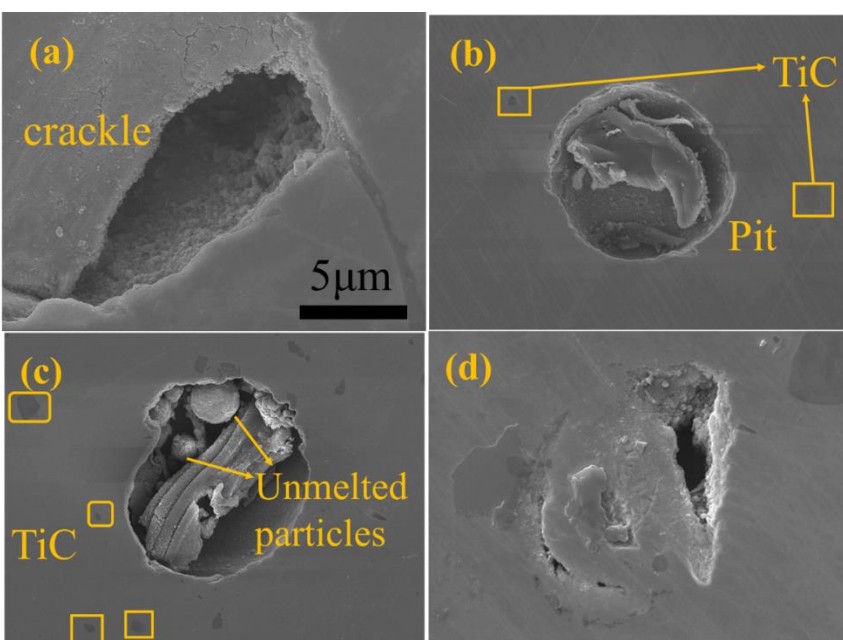

**Figure 5.** SEM images of the different defects on the sample surface: (**a**) Image of cracks containing TiC; (**b**)Image of a pore containing TiC; (**c**) Image of a hole containing unfused particles; (**d**) Typical SEM image of a defect.

### 3.2. Microstructure Characterization and Phase Analysis

Figure 6 shows the SEM images of the TiC/M2HSS composites with different TiC content. The organization of the LPBF-print sample with pure M2HSS was dominated by honeycomb isometric fine grains with a grain size of approximately 1–2 μm, as shown in Figure 6a, which was similar to the results reported by Liu et al. [8,9]. The organization of the TiC/M2HSS sample is shown in Figure 6b–d, which was dominated by ultra-fine isometric grains with a grain size that was almost always less than 1 μm, and the presence of grain boundaries, where the grain size was almost always less than 1 μm, with granular precipitates at the grain boundaries (enlarged image). The sample with 1% TiC showed a significant grain refinement compared to the original material; however, as the TiC content increased, the grains were not further refined.

The refinement could be attributed to the fast solidification process of the LPBF process, which resulted in a short grain growth time, the addition of TiC particles, which provided nucleation sites and provided a pegging effect, hindering grain growth, and the precipitates at the grain boundaries, which hindered grain growth [28], and acted as a refinement process.

Figure 7a shows the XRD pattern obtained in this experiment in the width range of 2θ (10° to 80°). Comparing the standard PDF card, the main phases of M2HSS prepared by LPBF were the FCC or BCC phases, and the phase of the MXCY-type carbides could be observed in the XRD pattern of the TiC/M2HSS composite with the addition of TiC. With an increasing amount of TiC, the intensity of the peaks of MXCY-type carbides also increased due to the precipitation of carbides induced by the addition of TiC [23].

According to Bragg's law, the 2θ angle between the X-rays and crystal plane was inversely related to the crystal plane spacing. Figure 7b focuses on intervals of 44°–45°, showing that as TiC content increased, the peak of BCC in the sample gradually moved to the right and the value of 2θ increased; thus, grain plane spacing d decreased and dislocations increased. This decrease in grain plane spacing was due to the aggregation and precipitation of TiC nanoparticles at the grain boundaries, which triggered lattice distortions and led to residual stress aggregation and increased dislocation density at the grain boundaries [17].

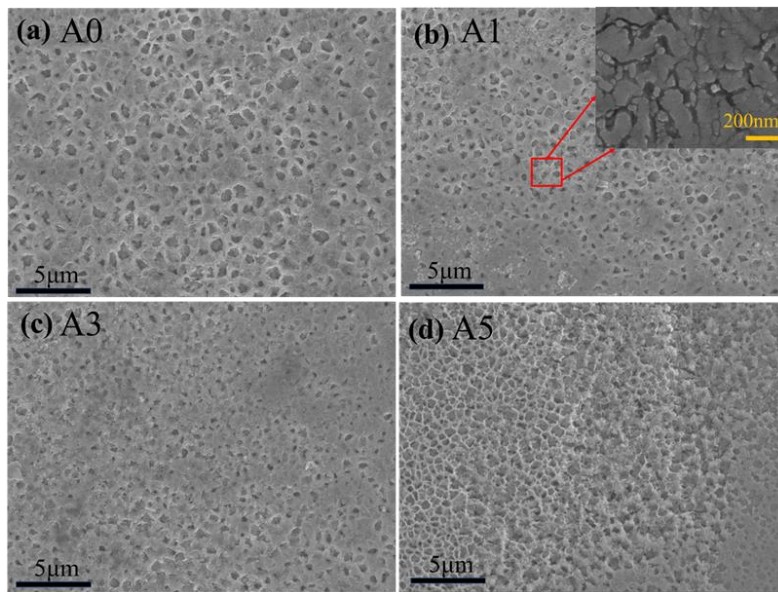

**Figure 6.** (**a**–**d**): SEM images of A0, A1, A3, A5 samples.

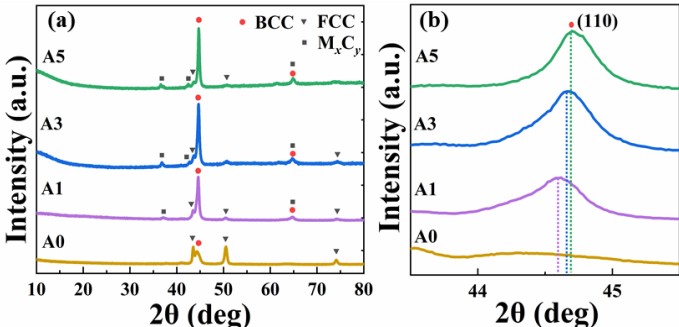

**Figure 7.** (**a**) XRD analysis of the A0, A1, A3, and A5 samples; (**b**) patterns at 41°–45° at a larger magnification.

To determine the effect of TiC addition on the stresses involved in the samples, all samples were characterized for residual stresses. Each sample was measured in three different areas of the surface and averaged. The test results for each sample are shown in Table 2.

**Table 2.** Samples residual stress.

| Sample Number | A0 | A1 | A3 | A5 |
|---|---|---|---|---|
| Stress value | 389.5 ± 29.6 MPa | −141.5 ± 10.2 MPa | −122.5 ± 10.9 MPa | −338.8 ± 32.6 MPa |

From the test results, it can be seen that the residual stress is 389.5 MPa when the sample is not added with TiC, at which time the residual stress is positive, i.e., perpendicular to the sample surface. When TiC was added, all the formed stresses became shear stresses, i.e., parallel to the sample surface. The residual stress of the sample with 1% TiC addition is −141.5 MPa, and the re-residual stress of the sample with 3% TiC addition is −122.5 MPa, which was slightly lower than that of the sample with 1% TiC addition, while the residual stress of the sample with 5% TiC addition is −338.8 MPa, which is a significant improvement compared with the samples of A1, A3. This is due to the instability of the melt pool caused by the addition of too many TiC particles. The proper addition of TiC helps the transition from positive stress to shear stress, while when 5% TiC is added it causes a large increase in the residual stress of the sample.

### 3.3. Hardness and Friction Properties

The Vickers hardness of the A0–A5 samples is shown in Figure 8a, where the Vickers hardness of the A0 sample without TiC was approximately 683 HV. This was because the rapid melting and solidification process during the LPBF process not only resulted in grain refinement, but also heated up the matrix portion of the sample during the layer-by-layer scanning process; thus, a high hardness could be obtained without heat treatment [41].

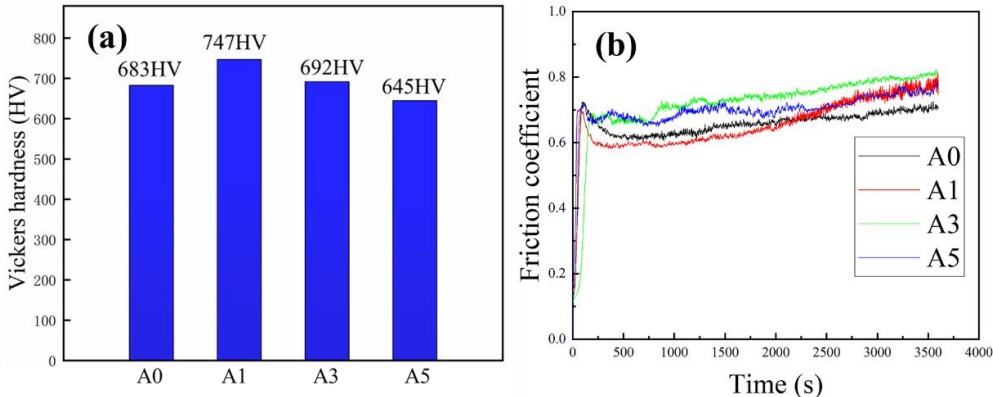

**Figure 8.** (**a**) Vickers hardness values of the A1, A2, A3, and A5 sample surfaces; (**b**) friction coefficient curve of M2HSS with different amounts of TiC after sliding for 60 min under 60 N.

The maximum Vickers microhardness of the TiC/M2HSS sample was 747 HV, and the A0 sample with 1% TiC addition reached peak hardness, with an increase of 9.37% over the pure M2HSS sample. The increase in hardness was due to the addition of TiC, which endowed the sample with a finer structure and higher energy absorption of the TiC particles, thus making the sample warmer during laser scanning. However, as TiC content increased, the Vickers hardness of the samples decreased, with Vickers microhardness measurements of 692 and 645 HV for samples A3 and A5, respectively. The decrease in hardness of the samples was due to the increase in surface defects such as cracks and holes due to the addition of excessive TiC (Figure 4).

Figure 8b shows the variations in the friction coefficient of the TiC/M2HSS composite with TiC content. The average friction coefficient of the samples decreased when 1% TiC was added and the time to reach the stabilization stage was slightly shorter. The friction factor was slightly higher when 3% and 5% TiC was added compared to the pure M2HSS samples. In general, the friction factor was related to the surface roughness of the samples [42], and the friction factor of the samples was slightly higher than the pure M2HSS samples due to the large number of defects such as pores and cracks on the surfaces of the A3 and A5 samples, in combination with the OM diagram shown in Figure 4.

In order to more accurately characterise the micro-zone hardness and modulus of elasticity of the material, the samples were tested using the nanoindentation method and the results are listed in Table 3. It can be seen from Table 3 that the micro-zone hardness of the samples follows the same trend as the macro hardness. This is due to the fact that the addition of TiC in the right amount can refine the structure and improve the hardness of the samples. However, when too much TiC is added, the defects in the sample increase and the hardness decreases. The modulus of elasticity of the samples also improves when 1% TiC is added due to the refinement of the structure, while it decreases when 3% and 5% TiC is added due to the increase in defects.

**Table 3.** Microzone hardness and microzone elastic modulus of different samples.

| Sample Number | A0 | A1 | A3 | A5 |
|---|---|---|---|---|
| Microzone hardness (GPa) | 5.66 | 7.17 | 6.30 | 6.02 |
| Microzone modulus of elasticity (GPa) | 158.33 | 163.89 | 140.54 | 102.47 |

The distribution of Figure 9a–d shows the shape of the 3D abrasion marks for samples A0, A1, A3, A5. The width and depth of the abrasion marks of sample A1 with 1% TiC addition were partially reduced, while the bottom of the abrasion marks of sample A0 without TiC addition was smoother and the bottom of the abrasion marks of sample A1 were rougher. This was due to the addition of TiC, which induced the precipitation and aggregation of carbide, and prevented contact between the grinding ball and sample matrix during the frictional wear process. Therefore, this improved the wear resistance and reduced the wear rate of the samples. The widths of the abrasion marks of the A3 and A5 samples with 3% and 5% addition were wider than the A0 samples. The wear volume of the A0–A5 samples is shown in Figure 9e. The wear volume of the samples without TiC addition was $3.1 \times 10^6$ μm$^3$. The A1 sample had the smallest volume loss, with approximately 39% less wear volume than the A0 sample, while the A3 sample had 16% more wear volume than the A0 sample, and the A5 sample had 35% more wear volume than the A0 sample. Therefore, adding too much TiC would lead to a decrease in the frictional wear performance of the samples, while adding the right amount of TiC would enhance the frictional wear performance of the samples and reduce the wear volume of the samples.

Figure 10a–d shows the color 2D plots of the abrasion marks on samples A0–A5, where the width of sample A1 with 1% TiC addition was clearly somewhat smaller than the widths of the other samples. The widest abrasion marks were found in sample A5 with 5% TiC addition, followed by sample A3. The middle sections of the sample abrasion marks were scanned using a laser scanning co-aggregation microscope, as shown by the arrows in Figure 9a–d, to obtain a two-dimensional profile of the abrasion marks. Figure 10e shows the two-dimensional view of the abrasion marks for samples A0–A5 in order from top to bottom, where A1 had the shallowest abrasion marks and sample A5 had the widest abrasion marks. By observing the contours of the abrasion marks, the bottom of the A0 sample without TiC addition was smooth and had no obvious bumps. The abrasion marks on samples A3 and A5 were deeper and sharper, with jagged bumps on both sides of the abrasion marks. This was due to an increase in internal defects in the samples due to the excess TiC, resulting in an increase in both the width and depth of the abrasion marks and a further increase in the wear volume.

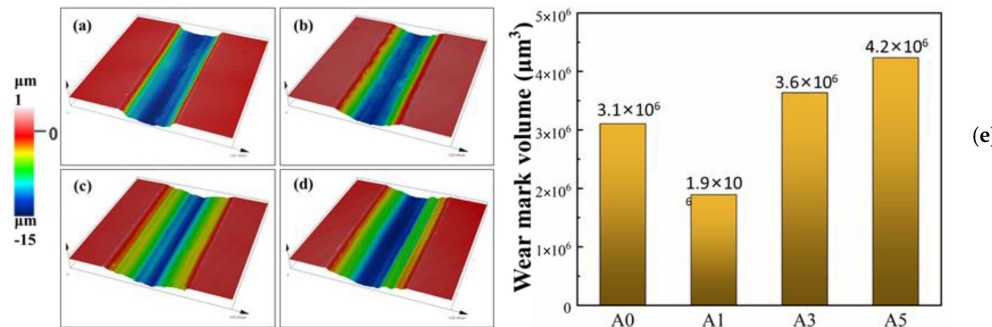

**Figure 9.** (**a**–**d**) The 3D surface morphologies of the wear marks on samples A0, A1, A3, and A5; (**e**) sample wear volume.

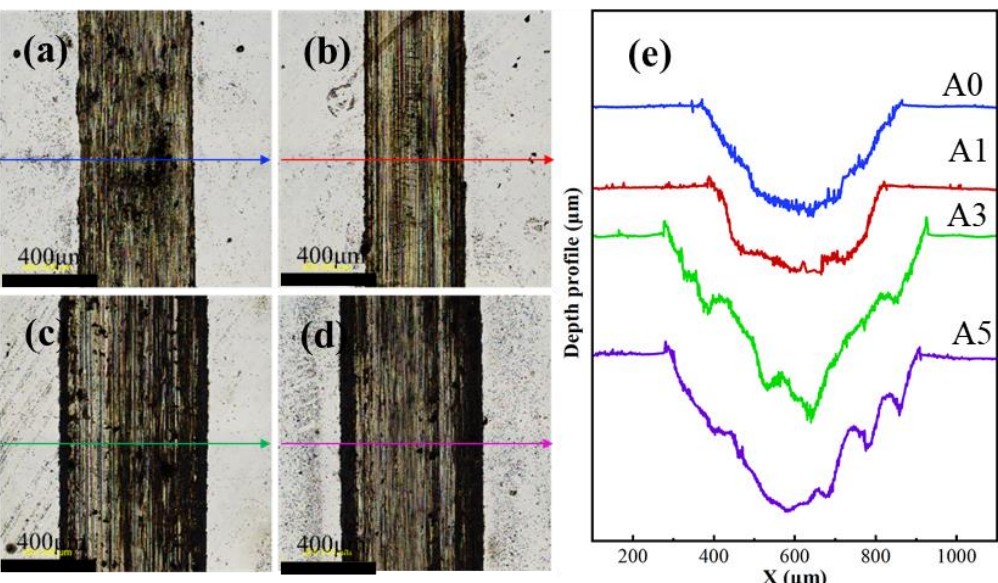

**Figure 10.** (**a**–**d**) Two-dimensional diagram of the sample wear marks; (**e**) cross-sections of the samples worn under 70 N for 60 min.

The microstructure diagrams of the A0–A5 sample abrasion marks are shown in Figure 11a–d, and Figure 10a shows the abrasion marks of the A0 sample of pure M2HSS. As shown in the figure, the abrasion marks on the surface of the sample were relatively wide with more scratches and pits. It was generally believed that the abrasion marks on the M2HSS sample were due to the plastic deformation of the grains, while the pits were caused by brittle cracking due to dislocation slip and stacking, which were more brittle and less plastic. The A1 sample with 1% TiC contained some dark grey TiC particles and cracks on the surface; however, the wear marks were narrower than those in the A0 sample, owing to the addition of TiC refining the microstructure of M2HSS and enhancing the plasticity of the sample. This resulted in fewer wear marks and aggregated TiC particles prevented the $Si_3N_4$ grinding balls from eroding the matrix. The A3 sample with 3% TiC had wider wear marks and a large number of pits and cracks, and the increase in defects within the sample allowed the grains to break away from the matrix during plastic deformation. The A5 sample with 5% TiC contained alternating black and grey abrasion marks, and a large number of pits were observed by magnifying some areas, which was due to the decrease in hardness of the sample. The process of grinding ball rubbing was caused by excessive defects in some areas of the sample surface, resulting in brittle cracking and consequent denting under the normal load of the grinding ball [43].

The addition of 3% and 5% TiC increased the internal defects of the sample and increased the surface cracks and porosity, which broke directly when the grinding ball was applied with a normal load and shear force, and the carbide particles were distributed within the matrix and acted as frictional substrates and participated in the frictional wear process. This increased the friction factor, making the volume of frictional loss larger and the wear marks wider and deeper. Therefore, the addition of TiC in the right amount refined the grains and precipitated carbides, enhancing the wear resistance of the sample. However, excess TiC addition would reduce the wear resistance of the sample.

The addition of 1% TiC helped to improve the frictional wear properties of the samples. The frictional wear mechanism of pure M2HSS with different TiC addition was described as follows. The wear resistance of the pure M2HSS material relied mainly on fine carbide grains such as WC, MoC, and VC, which were diffusely distributed in the matrix [44–53] with a grain size in the matrix ranging from 1 to 2 μm. When the grinding ball was applied to the sample surface with a normal load and shear force, the grains of M2HSS were pulled off, and the fragmentation is shown in Figure 12a. The addition of 1% TiC refined the grains

of M2HSS to less than 1 μm and dispersed the internal stresses. When the grinding ball applied a normal load and shear force to the surface of the sample, grain refinement better dispersed the forces and TiC precipitated and collected in the matrix, preventing further contact between the friction substrate and the substrate. Therefore, the M2HSS friction coefficient increased and the wear volume decreased, as shown in Figure 12b.

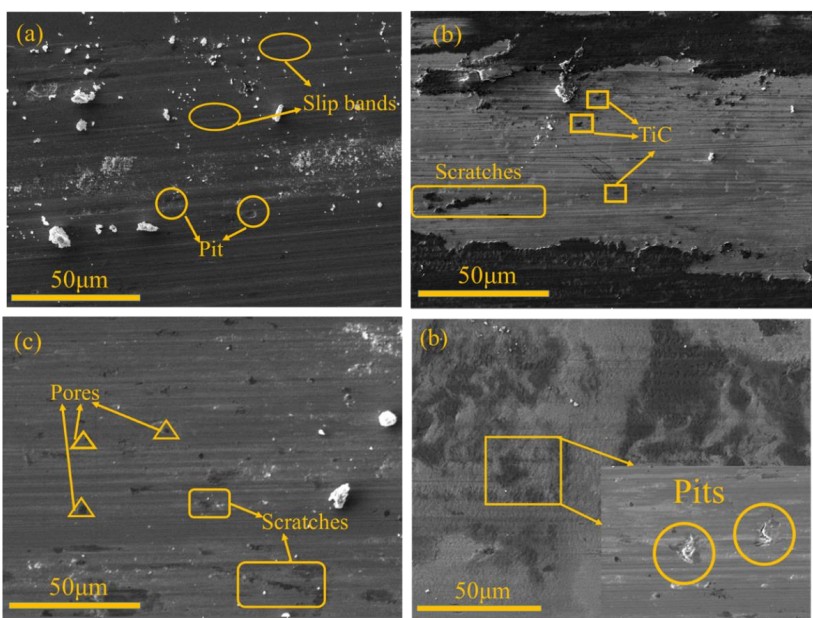

**Figure 11.** SEM images of the worn microstructures of the (**a**) A0, (**b**) A1, (**c**) A3, and (**d**) A5 samples.

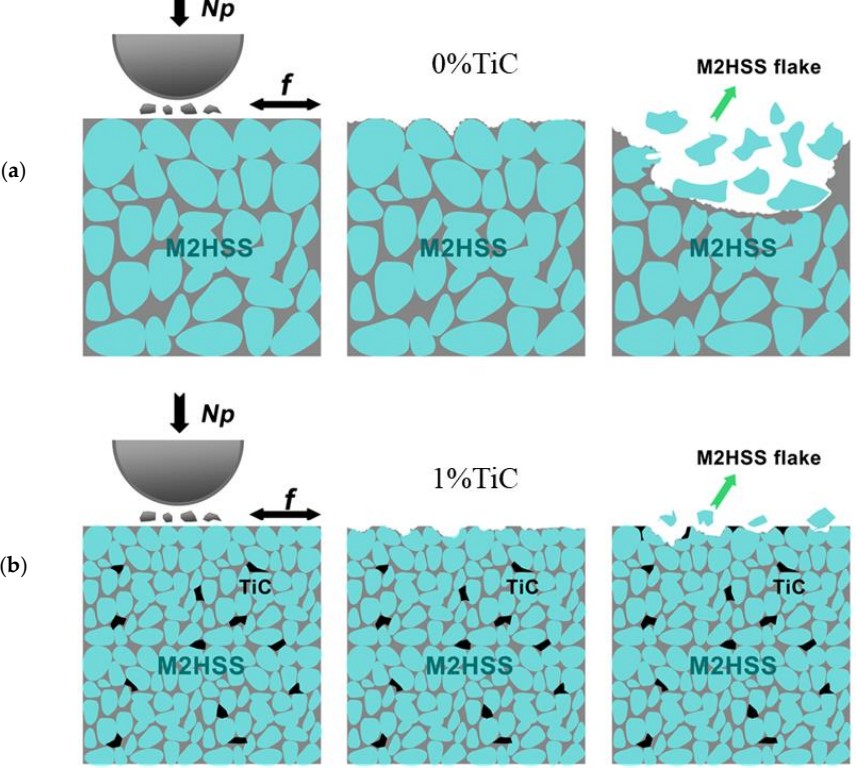

**Figure 12.** Schematic of the friction and wear mechanisms of the different alloys: (**a**) M2 HSS, (**b**) 1%TiC/M2HSS.

## 4. Conclusions

In this study, TiC/M2HSS composites were successfully prepared using physical mixing and LPBF processes. High density (97%) TiC/M2HSS samples were successfully prepared by investigating the relationship between the volumetric energy density, as well as the densities and surface morphologies of the samples. The effects of TiC content (0%, 1%, 3%, and 5%) on the microstructure, phase composition, defects, and frictional wear properties of the M2HSS samples were investigated. The following results were obtained.

1. The addition of TiC had a significant effect on the grain size of the TiC/M2HSS samples and promoted grain refinement of the samples.
2. The phase composition of the TiC/M2HSS samples consisted mainly of residual FCC and BCC phases and carbides, whereas the pure M2HSS samples consisted mainly of residual FCC and BCC phases.
3. As the TiC content increased, the hardness of the composite showed a tendency to increase and then decrease, and the maximum microhardness of the sample was 747 HV at 1% TiC content. Thus, the increase in hardness was mainly due to grain refinement.
4. The strengthening mechanism of the TiC/M2HSS samples was mainly the precipitation of carbide induced by TiC in the matrix, which prevented further erosion of the matrix by the grinding balls, and grain refinement also contributed to the frictional wear performance.

**Author Contributions:** Conceptualization, Y.L. and D.Z.; methodology, Y.L. (Yan Liu); software, Y.L. (Yan Liu); validation, Y.L. (Yan Liu), Y.L. (Yue Li) and D.Z.; formal analysis, Y.L. (Yan Liu); investigation, D.Z.; resources, Y.L. (Yan Liu); data curation, Y.L.(Yue Li); writing—original draft preparation, Y.L. (Yan Liu); writing—review and editing, D.Z.; visualization, Y.L. (Yan Liu); supervision, S.W.; project administration, D.Z.; funding acquisition, D.Z. All authors have read and agreed to the published version of the manuscript.

**Funding:** The work was supported by the National Natural Science Foundation of China (No. 52074128), Natural Science Foundation of Hebei (E2020209014, E2021209146).

**Institutional Review Board Statement:** Not applicable.

**Informed Consent Statement:** Not applicable.

**Data Availability Statement:** Not applicable.

**Conflicts of Interest:** The authors declare no conflict of interest.

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
