# Peer review of "Effects of Nano TiC on the Microhardness and Friction Properties of Laser Powder Bed Fusing Printed M2 High Speed Steel"

_coatings, doi:10.3390/coatings12060825_

Round 1

Reviewer 1 Report

The paper presents experimental results on to the microstructure and mechanical properties of TiC/M2HSS metal matrix composites samples prepared using LPBF process. It was found that the addition of 1% TiC had a positive effect on grain refinement, improvement of the frictional wear performance and increasing of microhardness.

The article is of interest of Coatings journal, but specific aspects mentioned in the following require the revision of the paper.

Overall, the text needs to be thoroughly revised much more rigorously since a lot of errors occurs and inconsistent terms are used, (e.g. “bulk energy density” in lines 124 and 125, “body energy density” in lines 119 and 120 and “volume energy density” in lines 132, 134 etc.).

Related to the Figure 3, it is not clear what "relative density" refers to, it should be explained.

Regarding to the parameters used in experiments, scanning interval (h) was not specified.

Line 199: “… melt pool solidification rate … extremely fast 10-6 - 10-7 k/s [36]”, if K represents Kelvin degree… this is not fast. It should be correlated with what is presented in line 40: “solidification occurring within 10-5 – 10-6 s [6]”

Line 224: figure 6b-d (and not 5b-d)

Figure 6: the TiC contents are not specified for each figure a) to d)

Figure 7: A0, A1, A3 and A5 (and not A1, A2, A3 and A4)

Figure 8a was not mentioned in the text.

Figure 9 is not clear as it does not specify for which sample the image is (e.g. figure 9a for sample A0). Figure 9b and figure 9c are probably reversed. On the other hand, wear mark volume related figure could be 9e (and not d)

Line 306: figure 10a-d (and not 9a-d)

Figure 10e: lacks a legend for sample identification
Figure 11: Image of the sample should be identified (e.g. A0 - figure 11a etc.…)

Figure 12b: The TiC percentage should be specified (as it is not the same at all percentage).

Some references are not referred in the text (e.g [7], [10]).

Reviewer 2 Report

The authors provide a paper dealing with the effect of Effects of nano TiC on the microhardness and friction properties of LPBF-printed M2HSS. The paper can be of interest for Coatings but MAJOR revisions are requested.

-          Do not use acronyms in the title. The same in the abstract M2HSS?, LPBF? What does it means?

-          The authors must comment in a more detailed way about the local mechanical properties of the materials. As a matter of facts, there are techniques which manage to extract fracture toughness, residual stress etc. on the local scale such as in doi.org/10.1016/j.matdes.2019.107762, doi.org/10.1016/j.matdes.2016.06.003. The authors must comment on these work and provide an improved discussion of the mechanical properties at the submicrometer scale. This I think will be of capital interest for the present paper, while representing a good outlook for the future work.

-          Connected to the previous point, a nanoindentaion analysis can also be performed especially in cartography mode to extract the elastic modulus and hardness for this composite materials. This can provide really high quality results.

Round 2

Reviewer 2 Report

-